# Gut Microbiota and Serum Metabolome in Elite Cross-Country Skiers: A Controlled Study

**DOI:** 10.3390/metabo12040335

**Published:** 2022-04-07

**Authors:** Jukka E. Hintikka, Eveliina Munukka, Maarit Valtonen, Raakel Luoto, Johanna K. Ihalainen, Teemu Kallonen, Matti Waris, Olli J. Heinonen, Olli Ruuskanen, Satu Pekkala

**Affiliations:** 1Faculty of Sport and Health Sciences, University of Jyvaskyla, 40014 Jyväskylä, Finland; johanna.k.ihalainen@jyu.fi (J.K.I.); satpekka@jyu.fi (S.P.); 2Turku Microbiome Biobank, Institute of Biomedicine, University of Turku, 20500 Turku, Finland; laevmu@utu.fi; 3Research Institute for Olympic Sports, 40700 Jyväskylä, Finland; maarit.valtonen@kihu.fi; 4Department of Pediatrics and Adolescent Medicine, Turku University Hospital, 20521 Turku, Finland; rajopi@utu.fi (R.L.); olli.ruuskanen@tyks.fi (O.R.); 5Clinical Microbiology, Turku University Hospital, 20521 Turku, Finland; tevika@utu.fi; 6Institute of Biomedicine, University of Turku, 20500 Turku, Finland; mwaris@utu.fi; 7Paavo Nurmi Centre, Department of Health and Physical Activity, University of Turku, 20540 Turku, Finland; olli.heinonen@utu.fi

**Keywords:** winter games, athletes, exercise, microbiology, metabolomics, lipids

## Abstract

Exercise has been shown to affect gut the microbiome and metabolic health, with athletes typically displaying a higher microbial diversity. However, research on the gut microbiota and systemic metabolism in elite athletes remains scarce. In this study, we compared the gut microbiota profiles and serum metabolome of national team cross-country skiers at the end of an exhausting training and competitive season to those of normally physically-active controls. The gut microbiota were analyzed using 16S rRNA amplicon sequencing. Serum metabolites were analyzed using nuclear magnetic resonance. Phylogenetic diversity and the abundance of several mucin-degrading gut microbial taxa, including *Akkermansia*, were lower in the athletes. The athletes had a healthier serum lipid profile than the controls, which was only partly explained by body mass index. *Butyricicoccus* associated positively with HDL cholesterol, HDL2 cholesterol and HDL particle size. The *Ruminococcus* *torques* group was less abundant in the athlete group and positively associated with total cholesterol and VLDL and LDL particles. We found the healthier lipid profile of elite athletes to co-occur with known health-beneficial gut microbes. Further studies should elucidate these links and whether athletes are prone to mucin depletion related microbial changes during the competitive season.

## 1. Introduction

Gut microbiota refer to the trillions of microbial cells inhabiting the gastrointestinal tract that, in addition to many other tasks, break down the macromolecules and nutrients from ingested food [1,2]. Bacterial metabolites such as short chain fatty acids (SCFAs) provide energy for muscles and intestinal epithelial cells [3,4,5]. Physical activity and lean body composition are associated with a gut microbiome that contains high abundances of health-promoting bacterial taxa [6,7,8,9], and higher microbial diversity is a common finding in athletes compared to sedentary controls [10]. Exercise may help develop a microbiome with a greater ability to harness energy from the diet and with an increased capacity for carbohydrate metabolism, cell structure, and nucleotide biosynthesis [11,12]. On the other hand, dysbiotic, i.e., metabolically unbalanced, gut microbiota have been shown to impair skeletal muscle adaptation to exercise [13]. Certain gut microbes have been shown to increase and diminish acutely in response to exercise [14], and high-intensity endurance training has even been linked to increased dysbiosis on some occasions [15].

During training and competition season, elite-level athletes are predisposed to increased levels of energy expenditure, stress and low-grade inflammation [16,17] and are at an increased risk of respiratory infections [18]. Chronic training can have notable effects on metabolic and inflammatory states [19], and these effects can persist in workloads below those constituting overtraining [20,21]. To this end, new studies suggest that certain microbial metabolic functions can enhance athletic performance and these functions could be accessed with certain bacterial taxa: Scheiman et al. [22] demonstrated that inoculation into mice of the lactate-fermenting species *Veillonella atypica,* isolated from Boston marathon competitors, increased the run times of the mice on a treadmill. Likewise, a recent similar trial on humans demonstrated that oral administration of a subspecies of *Bifidobacterium longum* isolated from a weightlifting Olympic athlete could increase Cooper’s running test results [23].

Both physical activity and the gut microbiome can have an instrumental role in maintaining cardiovascular and metabolic health [24,25,26,27], yet chronic high-intensity exercise can also increase inflammation and cause shifts in the microbiome. Integrative studies on the gut microbiome and the interplay of microbial and systemic metabolism in elite athletes are still scarce. We investigated whether the gut microbiota and serum metabolome of high-level cross-country skiers at the end of the competition season differs from those of age and sex-matched non-athletes.

## 2. Results

### 2.1. Gut Microbiota Diversity

The gut microbiota alpha-diversity measures, Chao1 (i.e., species richness, Figure 1A) and Shannon index (i.e., species diversity, Figure 1B) were found to be similar among the athletes and controls. However, the athletes had lower phylogenetic diversity than the controls (Figure 1C). According to the Bray–Curtis distance and PERMANOVA analysis (*p* = 0.66), the groups did not differ in beta-diversity, that is, in inter-individual species diversity of the gut microbiota (Figure 1D).

### 2.2. Gut Microbiota Composition

The average gut microbiota composition of the athletes and controls at the phylum, family and genus levels are shown in Figure 2. In the athletes, Bacteroidetes (50.4% of all sequences) and Firmicutes (46.0%) were the dominant bacterial phyla, followed by Proteobacteria (2.3%), Actinobacteria (0.79%), Verrucomicrobia (0.05%), Cyanobacteria (0.20%) and Tenericutes (0.03%). In the controls, Firmicutes (48.3%) and Bacteroidetes (46.2% of all sequences) were the dominant bacterial phyla, followed by Proteobacteria (3.36%), Actinobacteria (1.57%), Cyanobacteria (0.33%), Verrucomicrobia (0.20%), and Tenericutes (0.14%). A total of 27 families and 82 genera were identified in the athletes and controls. The family *Bacteroidaeae* (13.3%, mostly genus *Bacteroides*) explained the dominance of Bacteroidetes in the athletes, and the family *Lachnospiraceae* (25.5%, mostly genus *Blautia*) the dominance of Firmicutes in the controls.

In the ANOVA-like comparison, differences in the abundance of six gut microbiota genera were found between the groups (Figure 3). The athletes had a lower relative abundance of *Phascolarctobacterium*, *Lachnospiraceae* UCG-001, *Bacteroides*, *Lachnoclostridium* and the *Ruminococcus torques* group as well as a higher relative abundance of the *Eubacterium eligens* group than the controls (*p* < 0.038 for all). However, LEfSe analysis, that takes into account biological consistency and effect size, revealed the phylum *Actinobacteria* and the genera *Akkermansia*, *Bifidobacterium,* the *Prevotellaceae* NK3B31 group, *Alloprevotella, Flavonifractor, Ruminococcaceae* UCG 014 and the *Ruminococcaceae* NK4A214 group, to be more abundant in the controls.

In the control group, the amount of weekly exercise was associated positively with the family *Pasteurellaceae* and inversely with *Acidamidococcaceae* (Appendix A). The training load of the athletes was associated inversely with *Enterobacteriaceae*, *Bacteroidaceae* and *Veillonellaceae*. In addition, age was inversely correlated with *Tannerellaceae* and body mass index (BMI) with *Clostridium* sp. K4410.MGS-306.

### 2.3. Serum Metabolome

Overall, the athletes and controls were largely similar in their serum metabolome. Over the two first principal components, principal component analysis (PCA) showed no separation between the groups (Figure 4A). The supervised method, partial least squares-discriminant analysis (PLS-DA), reached a predictive ability of 0.75 (Q squared) and showed a moderate level of separation between the groups (Figure 4B). None of the metabolites differed by large factors (Figure 4C). A random forest classification task reached a sensitivity of 0.7 and a specificity of approximately 0.8 at peak (Figure 4D).

The age of the participants associated positively with circulating lipids including total cholesterol, total apolipoproteins, and LDL particle concentrations. BMI associated negatively with HDL cholesterol, average HDL size, amino acids, and glycolysis-related metabolites (Appendix A). Despite similarities between the groups, the athletes had higher total concentrations of both total HDL and its subfraction HDL2 compared to controls (Figure 5). Apolipoprotein A, the backbone of HDL, was higher and the mean HDL particle size was larger in the athlete group. In addition, the ratio of saturated fatty acids to total fatty acids was significantly elevated in the athletes. After adjusting for age and BMI, the group differences remained significant (see Appendix A).

The absolute and relative amounts of lipids within the different-sized HDL particles varied both within and between the groups. The athletes had a higher concentration of large and very large HDL particles (Figure 6A) and, consequently, higher lipid concentrations in these particles (Figure 6B). In the large and very large HDL particles (typically corresponding to HDL2 subfractions), the athletes had a higher cholesterol to phospholipid ratio than the controls did (Figure 6C). This difference was reversed as the particle size decreased.

The primary ketone bodies beta-hydroxybutyrate (bOHB) and acetoacetate were slightly lower in the athlete group, as was glycerol. In addition, the athletes had a higher concentration of pyruvate (Figure 7). After adjusting for age and BMI, the group differences in these metabolites remained significant (see Appendix A).

### 2.4. Associations between the Metabolites and Gut Microbiota

Notably, both microbial genera (Figure 8) and families (Figure 9) formed visible clusters according to their correlation coefficients with the lipoprotein lipid contents, with certain genera or families standing out. Most notably, a beneficial butyrate-producing genus, *Butyricicoccus*, positively associated with HDL and HDL2, as well as large to very large HDL concentrations and lipid contents (*p* < 0.05 for all). This genus also inversely associated with serum acetoacetate and albumin concentrations. *Collinsella* inversely associated with HDL and HDL2 concentrations and lipid concentrations in large HDL particles. The genus positively associated with medium to large VLDL concentrations and VLDL lipid contents. Both the *R. torques* group and *Lachnospiraceae* UCG-008 positively associated with the lipoprotein concentration and lipid content in the range of very small VLDLs to medium LDLs. Both genera also negatively associated with serum acetate.

At the family level (Figure 9), *Muribaculaceae* inversely associated with the lipoprotein concentration and lipid content in the range from large to small LDLs. They were also inversely related with virtually all circulating lipids. *Prevotellaceae* and *Coriobacteriaceae* associated with chylomicrons and large to medium VLDLs. In addition, *Prevotellaceae* associated with total triglycerides and tended to associate with triglycerides in almost all classes of lipoproteins, as apparent from the heatmap. *Christensenellaceae*, having only one representative genus, associated with medium sized HDLs. In addition, *Rikenellaceae* were found to inversely associate with histidine, branched chain amino acids (leucine, isoleucine, and valine), lactate, and pyruvate.

We ran a confirmatory analysis using multiple regression on the described associations between bacterial taxa and the metabolites (Appendix A). HDL2 cholesterol, HDL size and acetoacetate were significant predictors of the *Butyricicoccus* abundance (F = 82.88, *p* < 0.01, R^2^ = 0.854). Total cholesterol and acetate were significant predictors of the abundance of the *R. torques* group (103.7, *p* < 0.01, R^2^ = 0.830).

## 3. Discussion

In this study, we show that both the gut microbiota and the serum metabolome profiles of elite athletes were largely identical with their age- and sex-matched non-athletic controls. The phylogenetic diversity of the gut microbiota was higher in the control group. We also found several bacterial genera to differ between the groups. We observed that the athletes had higher serum concentrations of total HDL cholesterol and the subfraction HDL2. In addition, the athletes had a higher mean HDL particle size and more lipids contained in the largest HDLs.

A recent study explored the associations between the gut microbiota and plasma metabolites using the same methods [28]. The study also found that several microbial taxa are associated with the lipoprotein concentrations and lipid contents, especially VLDLs and HDLs. Our study and theirs describe only one intersecting taxon, the family *Christensenellaceae*, with somewhat similar associations found in both studies. This particular bacterial family is notable for having exceptional heritability and its abundance has been shown to be inversely associated with BMI and body fat [29,30], traits which usually mediate lipoprotein levels. Another recent study [31] also explored these associations using 16S rRNA gene sequencing and targeted metabolomics using liquid chromatography/mass spectrometry (LC-MS) rather than NMR, that was used here. It reported significant associations between BMI, branched chain amino acids and four microbial taxa (*Blautia*, *Dorea*, *Ruminococcus*, and SHA-98).

In addition, a set of studies has explored the gut microbiomes of professional rugby players compared to healthy controls [11,32]. The microbiome was both compositionally and functionally different between the groups and strongly associated with various aspects of diet. In contrast to our results, the alpha- and phylogenetic diversities of the gut microbiota were found to be higher and *Akkermansia*, a SCFA-producing and health-beneficial genus of the phylum *Verrucomicrobia,* more abundant in the athletes. The findings of the rugby study could partly be explained by dietary extremes observed in the groups, whereas we did not analyze diet. In another study comparing senior orienteers to sedentary controls, no differences in the microbial diversity were found altogether [33]. Previously, microbial diversity and the abundance of *Akkermansia* have been positively linked with physical activity in several other studies [6]. Since *Akkermansia* did not associate with weekly physical activity in either of our study groups, our results somewhat contradict these findings. However, it should be noted that our control group was not sedentary but exercised normally.

The genus *R. torques* group, which was more abundant in the controls than it was in the athletes (1.3% vs. 0.8%, respectively), has been associated with increased intestinal barrier leakage and increased serum triglycerides [34,35]. Although *R. torques* group did not associate with serum triglycerides in our study, we found the genus, along with *Lachnospiraceae* UCG-008, to be positively associated with total cholesterol, small VLDLs and large LDLs, which are mediators in hyperlipidemia [36,37]. These genera had an inverse association with serum acetate, which is both a microbial metabolite and a less ubiquitous ketone body. Interestingly, the species *R. torques*, along with *Akkermansia muciniphila* and species of *Bacteroides* and *Bifidobacterium,* is a consumer of mucins, the glycosylated proteins in the intestinal mucosa [38]. These genera were also less abundant in the athletes, which could indicate lower mucin availability, possibly due to high physical stress during the competitive season. However, this hypothesis warrants further studies.

*Butyricicoccus* is a butyrate-producing genus of the gut bacteria that has been proposed as a next generation probiotic [39]. The most prominent species of this genus, *B. pullicaecorum* has been shown to inversely associate with the incidences of ulcerative colitis and Crohn’s disease [40], suggesting that it can be health-beneficial. Ketone bodies acetoacetate and bOHB can be used as a substrate by several butyrate-producing bacteria [41,42] including *Butyricicoccus*. Here, the relative abundance of *Butyricicoccus* was not higher in the athlete group, but acetoacetate inversely and significantly predicted its abundance, which might indicate utilization of this ketone body by *Butyricicoccus*. Further, this genus was positively associated with higher HDL and HDL2 cholesterol and larger HDL particle size. This association has not been observed before but is an encouraging discovery considering future studies and raises the question whether the species belonging to this genus could mediate a healthier blood lipid profile when orally administered as probiotics or, vice versa, whether they thrive under such conditions.

The HDL cholesterol associates with improved health outcomes [43,44], and analyzing lipoprotein particle size and different HDL subfractions allows for more accurate probing of lipoproteins and associated health risks [45,46,47]. Mature HDL particles that are formed from small, lipid-poor, pre-beta HDLs are further divided into subfractions HDL2 and HDL3 according to their density and size [45]. Of the subfractions, HDL2 are larger, less dense and more lipid-rich forms of HDL. Both subfractions transport cholesterol into the liver to be excreted as bile salts, but HDL2 is also able to excrete cholesterol directly via transintestinal cholesterol excretion [44]. In our study, we found the athletes to have a higher proportion of cholesterol in the largest HDLs and, conversely, a lower proportion in the smallest ones. This could indicate more efficient cholesterol excretion in the athletes either as bile salts or through transintestinal cholesterol excretion. The gut microbes can both deconjugate bile salts and assimilate cholesterol [48], and some bacteria, including *R. Torques,* readily convert primary bile acids into secondary [49]. This could in part explain the observed associations between bacterial genera and lipoproteins.

A limitation of this study is the difficulty in selecting a control group matched for body composition. No body composition measurements were performed in the study and BMI alone can give a limited insight into this. Dietary data was not recorded in this study, and although not the primary interest in the study, the diet can affect both the gut microbiota and lipid parameters. Generally, elite athletes can be presumed to eat a different diet from the general population, e.g., a diet rich in carbohydrates and protein [50], while an individual diet can vary depending on many factors [51]. The absence of overnight fasting can confound the concentrations of some metabolic markers such as total triglycerides, yet this heterogeneity affected both groups equally. Since our study is cross-sectional, all results are associative and are interpreted as such. When comparing athletes and the general population, genetic background can confound some of the associations since the same polygenic risk factors can predict both physical activity and health outcomes [52,53]. Likewise, the gut microbiome exhibits at least some degree of heritability [54]. The possible ergogenic capacity (e.g., ability to ferment lactate) in the gut microbiome of elite skiers should be elucidated with further studies.

To conclude, we found lower phylogenetic diversity in elite cross-country skiers as well as minor overall differences in the gut microbiota composition and serum metabolome between the athletes and controls. The athletes had lower abundances of several bacterial genera, including many mucin-degrading bacteria, and a healthier serum lipid profile. *Butyricicoccus*, a genus with potential as a next-generation probiotic, was associated with higher HDL cholesterol and larger HDL particle size. In addition, we found the *R. torques* group, a genus associated with gastrointestinal disorders and dyslipidemia, less abundant in the athlete group, and associated with more abundant VLDL and LDL particles. Whether the athletes’ serum lipid profile is facilitated by the microbiome or vice versa, or whether these factors share a genetic makeup, could be studied with metagenomics and fecal metabolomics. Further longitudinal studies could also investigate whether elite athletes during competitive season are prone to mucin depletion and related changes in microbial composition and diversity.

## 4. Materials and Methods

### 4.1. Study Design and Population

This observational case-control study was carried out during the Finnish Nordic Ski Championships in Äänekoski, Finland between 28 March and 1 April 2019, an event held at the end of the skiers’ competition year. The study recruitment was accepted by 27 of 28 athletes belonging to the national Nordic Ski Team of Finland. The training season of the athletes had started at the beginning of May and the competition season at the beginning of November in 2018. The athletes had, therefore, experienced heavy physical stress for 11 months. For every athlete, one healthy, moderately exercising (<6 h per week) control subject was recruited from among the students and staff of Turku University Hospital and University of Turku, Finland. The controls (*n* = 27) were matched for age (+/− 2 years) and sex. The control subjects were studied in Turku, Finland according to the same study protocol as the athletes in Äänekoski, between 2 and 11 April 2019.

The clinical data and health-related information from the athletes and controls were collected by interview at the study visit by the study nurse (Table 1). The training load of the athletes was collected from the day-to-day training diary data of the previous 11 months.

The blood samples were collected by the study nurse a day before the competition. The athletes had followed their own individual training protocol when preparing for the following day’s race. The time frame between the last training bout and the blood sampling varied and we did not record the exact time. Due to the competition circumstances, no fasting from athletes and, correspondingly, nor from controls was required. The blood samples were centrifuged, and the serum separated immediately. The serum was aliquoted, immediately frozen at −20 °C and then stored at −80 °C.

### 4.2. Fecal Sample Collection and Microbial DNA Extraction

Both athletes and control subjects received a package containing self-collection equipment and instructions for fecal sampling. The package also included a questionnaire focusing on health status and lifestyle factors at the time of fecal sampling. The participants were guided to send both the sample and questionnaire to the Microbiome Biobank laboratory (University of Turku, Finland) by mail as soon as possible after the collection. The specimens were collected into OMNIgene^®^•GUT collection tubes (DNA Genotek, Kanata, OT, Canada) according to the manufacturer’s instructions. Briefly, the participants were guided to collect a small amount (approximately 500 mg) of fecal material into the tube, to homogenize the sample by vigorous shaking for 30 s and to mark the date and time of the sampling on the accompanying collection form. As OMNIgene^®^•GUT collection tubes include a stabilizing solution that guarantees DNA integrity in typical ambient temperature fluctuations and stability at room temperature for as long as 60 days, collection, storage and shipping of the samples could be performed at ambient temperatures [55].

At the laboratory, the samples were homogenized by gentle mixing, and the bacterial DNA was extracted from 200–250 µL of sample solution with a GXT Stool Extraction Kit VER 2.0 (Hain Lifescience GmbH, Nehren, Germany). Before the extraction, an additional homogenization by bead-beating in 1.4 mm Ceramic Bead Tubes (MO BIO Laboratories, Inc., Carlsbad, CA, USA) at 1000 rpm for 3 minutes with a MO BIO PowerLyzer™ 24 Bench Top Bead-Based Homogenizer (MO BIO Laboratories, Inc., Carlsbad, CA, USA) to enhance the cell lysis. The DNA concentrations were measured with a Qubit dsDNA HS Assay kit and Qubit 2.0 fluorometer (Thermo Fisher Scientific, Waltham, MA, USA), and the DNA samples were stored at −75 °C.

### 4.3. 16 S rRNA Gene Sequencing and Sequence Data Processing

The gut microbiota profiles were analyzed by 16S rRNA gene sequencing. To this end, the variable region V4 of the bacterial 16S rRNA gene was amplified with custom-designed dual-indexed primers and sequenced with the Illumina MiSeq system as described [56]. The raw 16S rRNA gene sequencing data were demultiplexed and the sequence adapters, primers and barcodes were clipped by using the Illumina BaseSpace platform. The raw sequence quality was checked with FastQC [57].

### 4.4. Gut Microbiota Composition Analyses

The 16S rRNA gene sequences were clustered to operational taxonomic units (OTUs) at 97% similarity using CLC Microbial Genomics Package (Qiagen, Hilden, Germany). The rRNA gene sequences were classified using the SILVA SSU Reference database (v132, 99%). The statistical analyses were performed with CLC Microbial Genomics Package. To analyze the gut microbiota alpha-diversity, Chao1 and Shannon indices were quantified. In addition, phylogenetic diversity was determined. The differences in the alpha-diversity measures between the groups were analyzed with the Kruskal–Wallis test. The beta-diversity analysis, which describes the dissimilarities in the ecosystem level community composition between samples, was based on Bray–Curtis distance and PERMANOVA for significance testing and the group differences were visualized with Principal Coordination Analysis (PCoA) in CLC. The taxonomic differences between the groups were analyzed with ANOVA-like comparison in CLC Microbial Genomics Package. The statistical significance in group comparisons was set at *p* < 0.05 after the multiple testing correction (Benjamini–Hochberg false discovery rate). The differences were also analyzed by LEfSe using the browser module and standard protocol [58]. The statistical significance was set at *p* < 0.05.

### 4.5. Metabolite Analyses

A high-throughput proton NMR metabolomics platform (Nightingale Health Ltd., Helsinki, Finland) was used to analyze the serum metabolic profiles as described earlier [59,60]. The analysis platform assesses 228 variables, including biomarkers of lipid and glucose metabolism, amino acids and ketone bodies. Zero values were imputed as the lowest values in the data set above zero, as per the laboratory instructions. Since more than half of the metabolites and derived parameters were not normally distributed, the Kruskal–Wallis test was used to test for feature-wise group differences. The effects of the background variables were investigated using Quade’s ANCOVA (nonparametric analysis of covariance). PCA and PLS-DA were used for multivariate analysis and visualization of the metabolites. To reduce the dimensionality in select visualizations, the metabolites were clustered and summed according to the biological function as reported by Nightingale Health.

For the correlation analyses between the microbes and metabolites, the microbial taxa with a prevalence of less than 10% at 0.001 relative abundance were excluded and the data was center log ratio (clr) transformed. Before the clr transformation, a pseudo-count of 10^−8^ was added to all fields to mitigate zero values. Any metabolites with missing values were also filtered out. The Spearman correlations between the metabolites and microbial taxa were measured and clustered on the microbial axis using the Ward method and Euclidean distance. The general linear model with ordinary least squares method was used to further test the associations. In addition, random forest and *XGBoost* in Python were used to test the predictive ability of the metabolites and taxa on the group membership.

## Figures and Tables

**Figure 1 metabolites-12-00335-f001:**
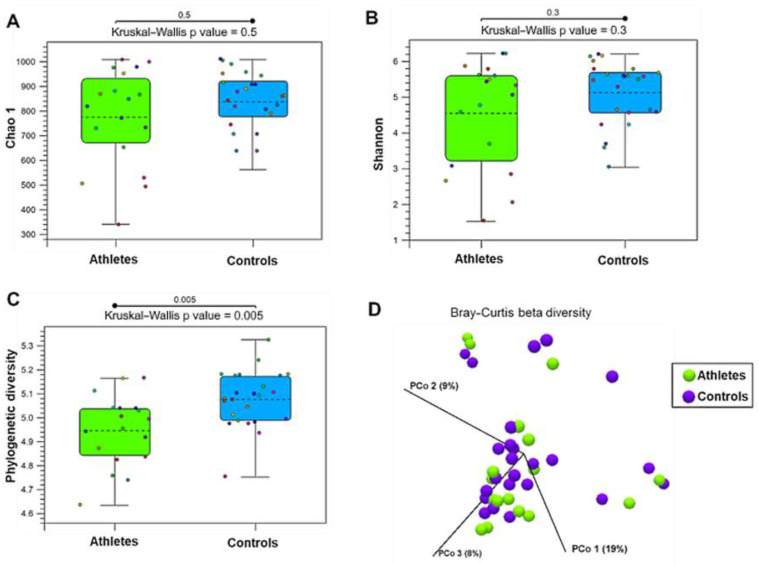
The diversity of the gut microbiota in the athletes (*n* = 27) and controls (*n* = 27). (**A**) Chao1, a measure of alpha-diversity indicating species richness. (**B**) Shannon index, a measure of alpha-diversity indicating species diversity. (**C**) Phylogenetic diversity. (**D**) Principal coordination (PCo) plot of beta-diversity analyzed by Bray–Curtis distance.

**Figure 2 metabolites-12-00335-f002:**
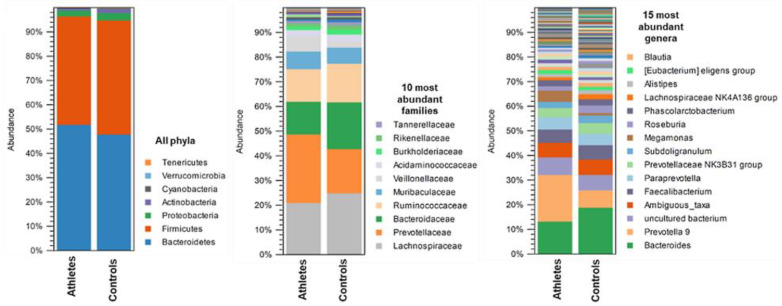
The average gut microbiota composition in the athletes (*n* = 27) and controls (*n* = 27) at phylum, family, and genus level.

**Figure 3 metabolites-12-00335-f003:**
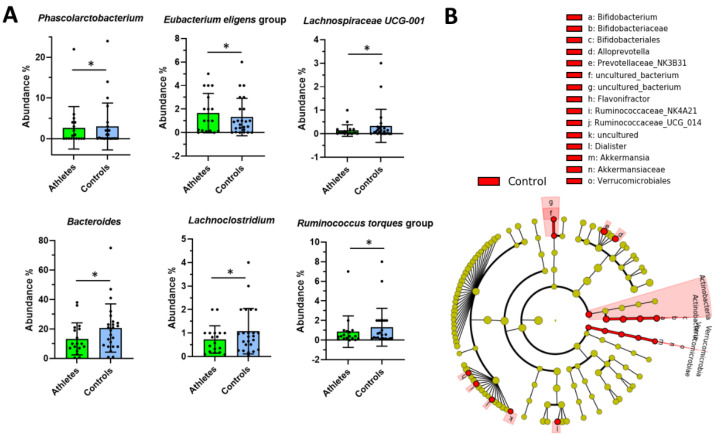
(**A**) The differential genus level abundances of the gut microbiota in the athletes (*n* = 18) and controls (*n* = 24). ANOVA-like comparison *p* values * < 0.05. (**B**) Cladogram from the LEfSe-analysis. The gut microbiota taxa that are highlighted were found to be enriched in the control group.

**Figure 4 metabolites-12-00335-f004:**
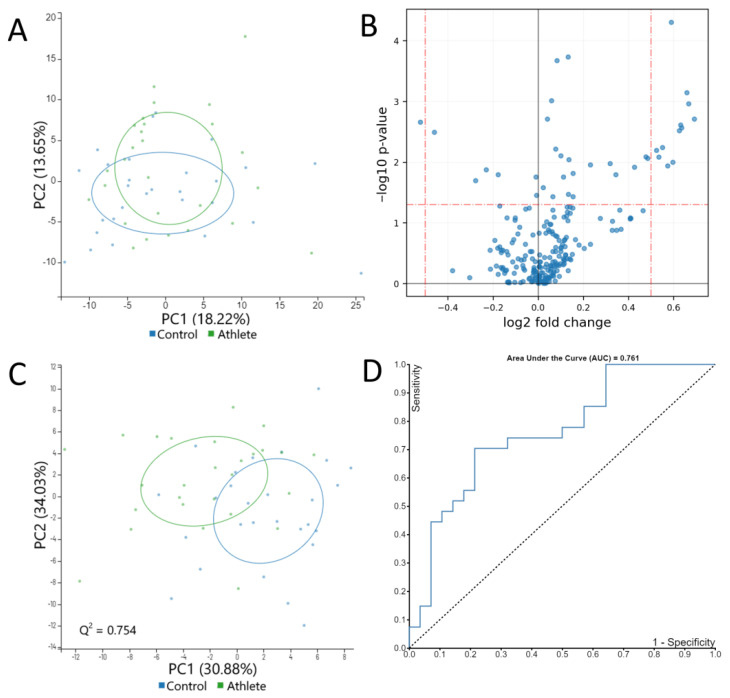
(**A**) The principal component (PC) analysis plot of all metabolites shows that most of the variance was explained by other factors than group. (**B**) The volcano plot indicates that the group differences were less than 2-fold for all metabolites. (**C**) The partial least squares-discriminant analysis plot and (**D**) receiver operand characteristics curve for random forest show that the classification task reached moderate accuracy.

**Figure 5 metabolites-12-00335-f005:**
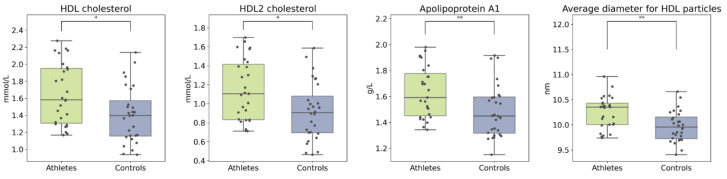
Boxplots of HDL cholesterol and related metabolites. Kruskal–Wallis *p* value: * < 0.05, ** < 0.01.

**Figure 6 metabolites-12-00335-f006:**
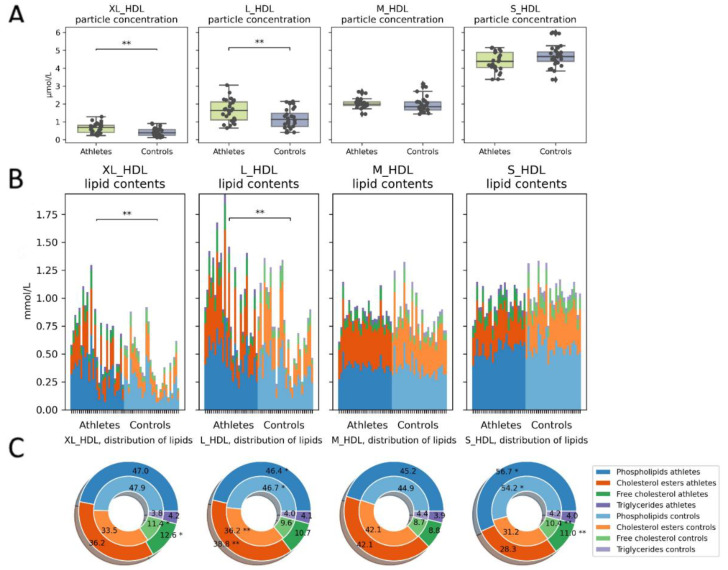
HDL particle concentrations and contents. (**A**) Total particle concentrations and (**B**) absolute lipid concentrations in the particles. Athletes had more large to very large HDL particles and consequently more lipids contained within them. (**C**) Relative proportions of lipids: the outer donut chart represents the athletes, and the inner represents controls. The athletes had a smaller phospholipid/cholesterol ratio in the large particles than the controls did. Kruskal–Wallis *p* value: * < 0.05, ** < 0.01.

**Figure 7 metabolites-12-00335-f007:**
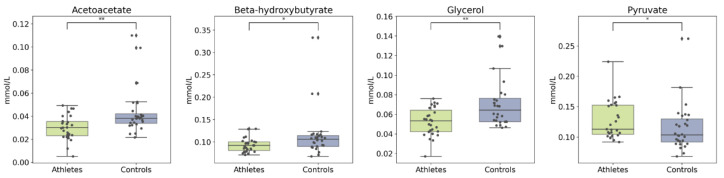
Boxplots of ketone bodies, glycerol, and pyruvate. Kruskal–Wallis *p* value: * < 0.05, ** < 0.01.

**Figure 8 metabolites-12-00335-f008:**
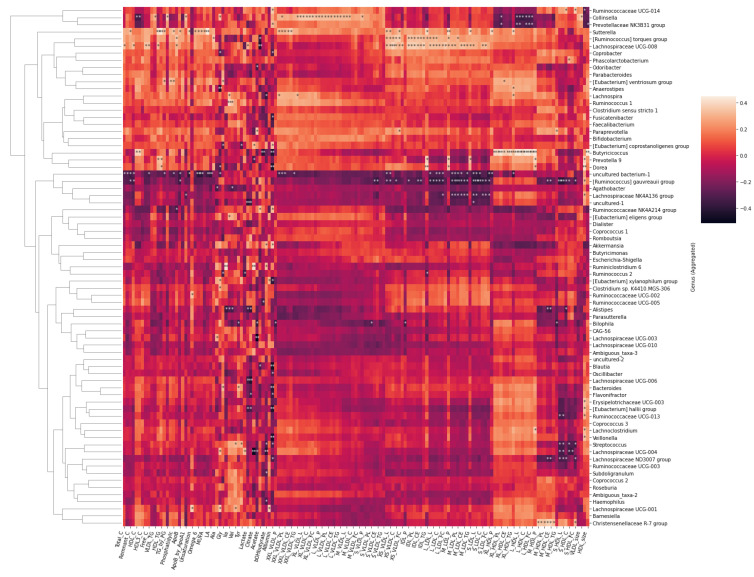
Heatmap of Spearman correlation coefficients between the serum metabolites and gut microbial genera. Spearman *p* value * < 0.05 ** < 0.01.

**Figure 9 metabolites-12-00335-f009:**
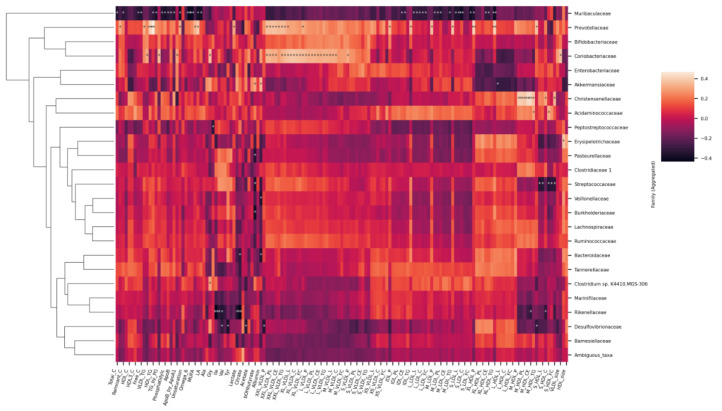
Heatmap of Spearman correlation coefficients between the serum metabolites and gut microbial families. Spearman *p* value * < 0.05 ** < 0.01.

**Table 1 metabolites-12-00335-t001:** Clinical characteristics of the study population. Values are represented as mean ± SD or as *n* of participants.

	Athletes (*n* = 27)	Controls (*n* = 27)
Age	27.1 ± 5.1	27.4 ± 5.6
Male/Female	14/13 (52/48%)	14/13 (52/48%)
BMI	22.05 ± 1.8	24.0 ± 3.5
Exercise load, h/week	15 ± 2	4 ± 1
Used oral antibiotics during previous 6 weeks	2	1

BMI = Body mass index.

## Data Availability

The data presented in this study are available in article and Appendix A.

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
