# Peer review of "Gut Microbiota and Serum Metabolome in Elite Cross-Country Skiers: A Controlled Study"

_metabolites, 2022, doi:10.3390/metabo12040335_

Round 1

Reviewer 1 Report

The present study compared the data from gut microbiota sequencing and metabolome between athletes and controls, which is interesting. However, some questions should be addressed.

The diet composition of each group should also be recorded and provided in this manuscript because diet contributes a lot to changing the gut microbiota and serum metabolites.

A LefSe analysis can be carried out to reveal the distinct genus of each group.

Fig.2, fig. 8 and fig. 9 are not at a high resolution, and clearer figures must be provided.

The discussion part is good, however authors should better elaborate more about the correlation between gut bacteria and lipid profile (line 216-231). Many spearman correlation analysis was made in the study, e.g. fig. 5, 7, 8, 9, and authors did not do the confirmation study, and only provided correlation analysis. More information about the correlation they found should better be provided.

Author Response

The present study compared the data from gut microbiota sequencing and metabolome between athletes and controls, which is interesting. However, some questions should be addressed.

The diet composition of each group should also be recorded and provided in this manuscript because diet contributes a lot to changing the gut microbiota and serum metabolites.

We thank the reviewer for the valuable comments on the manuscript. The diet composition, unfortunately, was not collected in this study, in case the data collection would have impeded normal training regime of the athletes. This has been discussed in the limitations section (264-278) in slightly more detail.

A LefSe analysis can be carried out to reveal the distinct genus of each group.

LefSe-analysis was carried out, as suggested, and a number of bacterial genera were enriched in the control group. Of these, we found the lower abundance of Akkermansia in athletes particularly interesting, since this mucin-degrading gut bacteria has been associated with physical activity before. Yet, our results on Akkermansia are contrary to some previous reports. The results have been described (4: 100-104) and further discussed (9: 213-224).

Fig.2, fig. 8 and fig. 9 are not at a high resolution, and clearer figures must be provided.

We apologize for the low resolution. The requested figures have been updated to higher resolution and should facilitate zooming in on the labels and the data points.

The discussion part is good, however authors should better elaborate more about the correlation between gut bacteria and lipid profile (line 216-231). Many spearman correlation analysis was made in the study, e.g. fig. 5, 7, 8, 9, and authors did not do the confirmation study, and only provided correlation analysis. More information about the correlation they found should better be provided.

We thank the reviewer for pointing out the issue. We ran regression analyses for the bacterial genera and metabolites described to make the associations more robust. We found HDL2, HDL size and acetoacetate to predict Butyricicoccus abundance, and total cholesterol and acetate to predict R. torques group abundance. The results have been described in the main text (8: 188-) and considered in the discussion.

Reviewer 2 Report

The authors present their findings comparing the gut microbiota and serum metabolome between elite cross-country skiers and those with moderate workouts. The authors look to fill the gap in knowledge concerning the gut microbiome and microbial metabolites in athletes. While the metabolome showed few differences between groups, there seemed to be a reduction in microbial diversity in elite athletes compared to controls. Overall, the authors provide associations between microbes and metabolites, with a particular focus on lipid profiles.

I commend the research efforts of the authors, as tying together the microbiome and metabolome of a system can be challenging. I found the microbiome work itself to be quite interesting and presented very well. In diseased state, literature seems to trend towards a greater diversity in microbiome as a healthier state. Here I wonder if the loss of diversity is necessarily a good thing. Are the athletes compromising another aspect of themselves due to these changes? The authors mention at the beginning (Ln 64) about increasing understanding of benefits and caveats, but it did not seem clear to me which findings were beneficial.

The author’s mention “microbial metabolite(s)” (Ln 63, 230) a few times throughout the manuscript. I found this to be somewhat misleading as it makes it appear that these metabolites are specifically microbe derived. One of the challenges of this type of research is that metabolites are involved in all aspects of the host. There is a constant exchange between microbes and host cells, as well as between the host and microbes themselves. Without some sort of labelling it is hard to say where these metabolites are derived and whether or not they are actually related to the microbes or the host. Is it possible that the limited metabolic changes that are seen are due to host derived changes that resulted in a switch in their metabolism? Some discussion on this topic would be well placed.

In the results section, the associations to BMI in figure 5 (Ln130) are not well described. Later on at Ln 149 there may be reference to it, but it is unclear. Please clarify and put result descriptions closer to corresponding figures.

Ln 221-3 what makes the positive correlation between Butyricicoccus and HDL/HDL2 cholesterol and HDL particles encouraging?

The addition of a non-exercise control (or stagnant) may have also be useful. If the microbiome and metabolic changes are truly linked based on activity level, this control group may have provided significant information. Should be considered as a follow-up or in future studies.

Minor

  • Abstract – italicize bacterial species
  • Figure 5 and 6
    • Please fix units so that the l for liters is capitalized (e.g. mmol/l should be mmol/L)
    • 2nd row, 1st box in Figure 5 has the y-axi title clipped
  • Figure 7: first box of each row has the y-axi title clipped
  • In supplemental, adjust “skier” to “athletes” to be consistent with other figures

Overall, it is difficult to assign metabolic shifts to microbiome networks. I think the authors have done the best that they can with the information given; but the associations shouldn’t be taken too definitive and the study would require further investigation.

Author Response

The authors present their findings comparing the gut microbiota and serum metabolome between elite cross-country skiers and those with moderate workouts. The authors look to fill the gap in knowledge concerning the gut microbiome and microbial metabolites in athletes. While the metabolome showed few differences between groups, there seemed to be a reduction in microbial diversity in elite athletes compared to controls. Overall, the authors provide associations between microbes and metabolites, with a particular focus on lipid profiles.

I commend the research efforts of the authors, as tying together the microbiome and metabolome of a system can be challenging. I found the microbiome work itself to be quite interesting and presented very well. In diseased state, literature seems to trend towards a greater diversity in microbiome as a healthier state. Here I wonder if the loss of diversity is necessarily a good thing. Are the athletes compromising another aspect of themselves due to these changes? The authors mention at the beginning (Ln 64) about increasing understanding of benefits and caveats, but it did not seem clear to me which findings were beneficial.

We thank the reviewer for the insightful comments and discussion on the results. Regarding the benefits and caveats, you pose the right question, however, we refrained from speculating too much whether the lower diversity is due to dysbiosis since it cannot  be directly inferred. The intro (page 2: lines 61-67) and the discussion (page 9: 213-224) have been hopefully made more concise on this issue.

The author’s mention “microbial metabolite(s)” (Ln 63, 230) a few times throughout the manuscript. I found this to be somewhat misleading as it makes it appear that these metabolites are specifically microbe derived. One of the challenges of this type of research is that metabolites are involved in all aspects of the host. There is a constant exchange between microbes and host cells, as well as between the host and microbes themselves. Without some sort of labelling it is hard to say where these metabolites are derived and whether or not they are actually related to the microbes or the host. Is it possible that the limited metabolic changes that are seen are due to host derived changes that resulted in a switch in their metabolism? Some discussion on this topic would be well placed.

Regarding the term “microbial metabolites”, we agree on the ambiguity of the term in this context. The metabolites studied here are mostly endogenous, with secondary microbial origins for some, such as acetate. In addition, compounds such amino acids and ketone bodies can be readily degraded by the gut microbes. Such interactions are difficult to capture in a cross-sectional study and for this reason we have chosen to use the phrase “microbiota and systemic metabolism” where applicable (abstract and 2: 64).

In the results section, the associations to BMI in figure 5 (Ln130) are not well described. Later on at Ln 149 there may be reference to it, but it is unclear. Please clarify and put result descriptions closer to corresponding figures.

We thank the reviewer for noting this. The associations with BMI were not of particular interest in this study and for this reason, the figures have been reduced on this part. Instead, we describe the group differences after controlling for age and BMI (page 6: 136 and 7: 157).

Ln 221-3 what makes the positive correlation between Butyricicoccus and HDL/HDL2 cholesterol and HDL particles encouraging?

The associations of Butyricicoccus and HDL are a novel finding and could warrant further studies on whether species of this genus could mediate healthier blood lipid profile, or vice versa. The discussion has hopefully been improved on this part page (9: 237-249).

The addition of a non-exercise control (or stagnant) may have also be useful. If the microbiome and metabolic changes are truly linked based on activity level, this control group may have provided significant information. Should be considered as a follow-up or in future studies.

We agree with the reviewer. Indeed, a sedentary control group would be invaluable in a setting like this, as long as the group can be also matched for confounders such as BMI. This is a good point and will be considered in future studies.

Minor

  • Abstract – italicize bacterial species
  • Figure 5 and 6
    • Please fix units so that the l for liters is capitalized (e.g. mmol/l should be mmol/L)
    • 2nd row, 1st box in Figure 5 has the y-axi title clipped
  • Figure 7: first box of each row has the y-axi title clipped
  • In supplemental, adjust “skier” to “athletes” to be consistent with other figures

Overall, it is difficult to assign metabolic shifts to microbiome networks. I think the authors have done the best that they can with the information given; but the associations shouldn’t be taken too definitive and the study would require further investigation.

The minor fixes requested have been implemented. Also, regarding the microbe-metabolite associations, we have hopefully made them more robust by running regression analyses on the bacterial taxa and metabolites of interest (8: 188-192).

Reviewer 3 Report

In my opinion, the study and results discussion are straightforward and clear to readers. Although the study population is limited, it can serve as a good reference for further study on the impact of strenuous exercise on the microbiome.

I think the manuscript can be accepted in it's current form.

Author Response

In my opinion, the study and results discussion are straightforward and clear to readers. Although the study population is limited, it can serve as a good reference for further study on the impact of strenuous exercise on the microbiome.

I think the manuscript can be accepted in it's current form.

We thank the reviewer for the kind words